# The Impact of *Constrictotermes cyphergaster* (Termitidae: Nasutitermitinae) Termites on Semiarid Ecosystems in Brazil: A Review of Current Research

**DOI:** 10.3390/insects13080704

**Published:** 2022-08-05

**Authors:** Mário Herculano de Oliveira, Arleu Barbosa Viana-Junior, Maria do Socorro Lacerda Rolim, Igor Eloi, Marllon Rinaldo de Lima Andrade, José João Lelis Leal de Souza, Maria Avany Bezerra-Gusmão

**Affiliations:** 1Programa de Pós-Graduação em Ecologia e Conservação, Universidade Estadual da Paraíba, Campina Grande 58429-500, PB, Brazil; 2Laboratório de Ecologia de Invertebrados, Programa de Pós-Graduação em Biodiversidade e Evolução, Coordenação de Zoologia, Museu Paraense Emilio Goeldi, Belem 66077-530, PA, Brazil; 3Programa de Pós-Graduação em Psicobiologia, Departamento de Fisiologia e Comportamento, Universidade Federal do Rio Grande do Norte, Natal 59078-970, RN, Brazil; 4Departamento de Solos, Universidade Federal de Viçosa, Viçosa 36570-900, MG, Brazil

**Keywords:** Isoptera, Blattaria, review, tropical forests

## Abstract

**Simple Summary:**

We present a systematic review of the biology of the termite *Constrictotermes cyphergaster* and its relationships with guests in South America, addressing issues concerning the research knowledge that has been accumulated and the gaps that still need to be filled regarding this termite, as well as its relationships with other organisms and with the environment.

**Abstract:**

Termites have global distributions and play important roles in most ecosystems, often with high nest densities and interesting associations with other organisms. *Constrictotermes cyphergaster*, is a termite endemic to South America, widely distributed and very conspicuous, and has therefore been considered a good model for filling in gaps in general termite ecology and their relationships with other organisms (e.g., termitophily). A systematic review (content and bibliometric analyses) was used to gather all published scientific knowledge related to *C. cyphergaster* as well as to observe trends, verify gaps, and direct new perspectives for future studies of this species. We identified 54 studies, of which more than 50% were published in the last five years (28 articles). The majority of the articles investigated the relationships between *C. cyphergaster* and macroorganisms (44.4%), followed by specific aspects of its biology (25.9%). The collaboration network revealed that links between researchers are still limited and modular, but trending topics have changed over time. Additionally, there are differences in the aims of the studies being carried out in the Caatinga and Cerrado domains, with some information focusing only on one of those environments. Our results show that some gaps in the biology and ecology of *C. cyphergaster* remain to be explored, although collaborative efforts between researchers open opportunities for suggesting future studies that would make relevant contributions to the general knowledge of termites.

## 1. Introduction

Termites are regarded as important ecosystem engineers in tropical habitats due to the significant environmental alterations they promote. Their greatest contribution to ecosystem functionality is through decomposition promoted by different feeding groups that consume wood, grass, litter, and soil organic matter [1]. Those insects contribute to the release of immobilized N, P, and carbon dioxide [2,3], as well as the increased capacity for cation exchange and leaching reversal [4,5]. They are also fast-responding and easily observed indicators of habitat quality [6], and can mitigate the effects of drought by maintaining ecosystem functions in tropical rainforests [7]. The influence of termites on semiarid soils is poorly known; however, this is of increasing interest in studies designed to determine their impacts. Termites are one of the few groups of macroinvertebrates found in semiarid and arid tropical savannas that do not experience estivation (developmental torpor or dormancy) during dry periods, thus maintaining their ecosystem functions [8,9]. Termites may even be the most important consumers in tropical dryland semiarid ecosystems and affect hydrological and nutrient cycling [10,11].

A large portion of the South American territory is classified as seasonally dry tropical forests-SDTF [12], with a distribution pattern known as the Pleistocene Arc [12,13]. Although South America holds over six hundred termite species [14], most research efforts have targeted only a few taxa, such as *Constrictotermes cyphergaster* (Termitidae:Nasutitermitinae) (Silvestri, 1901). This species has been recorded in many countries throughout South America, including Argentina, Bolivia, Paraguay, and Brazil [15]. It is the single most abundant and conspicuous nest-building nasute termite in the drylands of Northeastern Brazil, with a high average nest density of 58/ha [16]; there are also very representative numbers of nests in the Brazilian Savanna (Cerrado) [17]. Their nests are constructed from a mixture of saliva and soil particles [18].

Why is a review focusing on a single termite species pertinent, however? Studies focusing on *C. cyphergaster* have encompassed a great deal of information concerning the ecological importance of the species, as well as its association with other organisms and the environment. The species has important influences on the semiarid areas of Brazil. In Caatinga shrub vegetation, for example, the nests of *C. cyphergaster* (Figure 1) contribute to the ecosystem dynamics of carbon cycling and provide refuge and mating lanes for many invertebrate species (some of which are only found inside the nests of *C. cyphergaster*), and even vertebrates, such as birds and reptiles [6,15,18,19,20]. As places with internal levels of humidity and temperature contrasting strongly with the dry semiarid climate, their nests also provide support for microorganisms, such as lichens and fungi [21,22]. Morphologically, *C. cyphergaster* has dimorphic workers, with a division of labor during foraging [23]; furthermore, the apterous lineages of this termite do not display sexual polymorphisms, as both soldiers and workers are male [24]. The species demonstrates preferences for certain plants as support for their nests, and they use some of the metabolites produced by those plants as defensive compounds [25,26].

As *C. cyphergaster* has been the target of several published studies (since the mid-1980s), there has been a growing interest in compiling the literature on this species. Furthermore, its ecological impacts on the environments where it is found, as well as its use as a research subject in various studies, have helped to answer a range of questions concerning the ecology of termites, particularly the Nasutitermitinae, making the information gathered here a reasonable estimate of the current state of knowledge of that species and a starting point for future research. Our goal was therefore to undertake a systematic review of the literature regarding *C. cyphergaster* and address two main questions: (i) which categories of knowledge have been the most frequently produced concerning *C. cyphergaster,* and what are the gaps in our knowledge; and (ii) whether the scientific community that studies *C. cyphergaster* isolated or well-integrated and collaborative. This study is thus directed at discussing the accumulated scientific knowledge concerning *C. cyphergaster,* indicating possible gaps in our understanding of their natural history, observing trends, and raising new questions for future research.

## 2. Materials and Methods

We used three databases (Web of Science, Scopus, and Scielo) to search for papers concerning *C. cyphergaster* (in September 2021). The following search terms were used in consultations of all three databases:

Terms of Search = “*Constrictotermes cyphergaster*” OR “*C. cyphergaster*” OR “*C cyphergaster*” OR “*Constrictotermes*” AND “*cyphergaster*”. The searches generated a total of 109 results. The exclusion of duplicates and screened papers resulted in a final total of 52 studies. Two additional studies, published after September 2021, were later added, thus resulting in a total of 54 works included in our synthesis (Appendix A). The texts were analyzed in two steps for this systematic review: a scientometrics analysis and an analysis of content. 

We used R software to conduct the bibliometric analysis and build graphics for the scientometrics analysis [27]. The information provided in the articles were downloaded as Bibtex format files and converted to the R data frame using the ‘convert2df’ function. The collaboration networks of the authors were built considering authors as levels and the cluster method ‘edge betweenness’, using the Bibliometrix package [28]. All the papers were downloaded and read, and a scientometrics matrix was built to analyze their contents.

All manuscripts were carefully studied to analyze their content, and some topic information has been extracted from them:

1. Works that generated information about the host or guests, or both. In this topic, we considered *C. cyphergaster* the host. All other invertebrate organisms were considered colony guests. A subdivision was created considering Inquilines as termite species other than the builders but living within the nests; the other invertebrate groups were considered termitophiles, as used by [29] for Staphylinidae beetles.

2. A general division with six levels:
Relationships with the environment: all papers directed towards the study of the modifications that termites made on the environment with their nests, and when the external environment influenced termites in their building location or behavior;Methods: studies aimed only at generating results for methods of maintenance of *C. cyphergaster* in bioassays or nest volume determination methods;Guest Biology/Evolution: Studies that focused on guest biology, or the evolution, phylogeny, or anatomy of guests of *C. cyphergaster*;Species/Specific aspects: Studies of specific characteristics of *C. cyphergaster* anatomy, morphology, the histology of individuals, or population size;Relationships with Microorganisms: Studies providing information on the relationships of *C. cyphergaster* with bacteria, lichens (mycobionts+ photobionts), and fungi;Relationships with Macroorganisms: works that focused on interactions with invertebrates or plants.

3. Another division targeted study sites, that is, whether the studies were conducted in the laboratory, in the field, or both. These studies were again separated according to the sampling spaces described by the authors in the Caatinga, Cerrado, or undescribed.

## 3. Results

Upon analyzing the 54 articles identified, we perceived a progressive increase in the number of studies focusing on *C. cyphergaster* as a model species, indicating an increasing interest in that termite. The number of published papers (28) in the last five years (2016–2021) was over twofold of that produced (11) between 2010 and 2015 (Figure 2). While the journals at the top of the list are specific journals (with social insect-focused scopes), not only has the number of papers increased as a function of time but also the impact factor (IF) of journals that authors published on the species. The journal with the highest impact factor that published information concerning *C. cyphergaster* had an IF = 7.963 [26]. Most of the studies that generated information related to this termite were carried out under laboratory conditions (Table 1).

The authors’ collaboration networks showed 17 clusters with links between them (Figure 3A), with the top authors (in terms of numbers of publications) being at the centers of the clusters. Inspections of the top authors’ productions indicated that the recent integration of new authors influenced production rates during the last ten years, mainly in the last five years (Figure 3B), and the objectives of those new authors likely influenced trending topics. The frequency of keywords in the studies has evolved, with terms such as “symbiosis”, “inquilinism”, and “cohabitation” appearing more frequently in the last two years (Figure 4).

The analysis of content lead to the division of the published texts in six categories (Figure 5A). The category with the most articles produced was ‘Relationships with Macroorganisms’ (*n* = 24). Among the publications in that category were papers focusing on bees, beetles, termites, wasps, other invertebrates, and plants associated with *C. cyphergaster* nests. The second most prolific category was ‘Species/Specific aspects’ (*n* = 14). Those categories together accounted for more than 70% of the published studies. The majority of works regarding guests targeted termitophiles, with a production very similar to that of investigations of inquilines (Figure 5B).

## 4. Discussion

Examinations of the general results indicated a tendency to maintain publications in specific insect field journal lines, although one would expect increased publications in more generalized journals with higher impact factors in light of the importance of new research targets to increase impact factors and the numbers of published studies, as well as the involvement of authors with new perspectives.

It is important to highlight that, aspects related to species-specific characteristics need to be part of further studies, such as studies with genetic information. Being a species with a wide distribution in Brazilian territory, mainly the Caatinga and Cerrado domains [17], its genetic profile, behavior, and morphology will reflect spatial variations according to varying environmental conditions. One of these aspects is worker dimorphism, which has been closely examined until now only through morphometric and behavioral studies of Caatinga populations [23]. Another aspect that currently displays a biome research bias towards Cerrado is the influence of soil characteristics on nest-building behavior [25].

There is growing interest in the genetic structures of termite populations [30], biogeography, and phylogeny, with recent studies of *Heterotermes* and *Kalotermes* [31,32]. Inbreeding is an integral part of the colony life cycle in many species, and has been the focal point in discussions of the evolution of eusociality in the group [33]. Nestmate recognition has been studied in *C. cyphergaster* using chemical analyses and behavioral parameters [34,35,36], but genetic studies that define the genetic structure of a colony have yet to be conducted, although that information is critical for studies of the biology of social insects [37].

Genetic studies would be useful for determining rather obscure traits of their biology—such as their interactions with microorganisms. Very little is currently known regarding the bacteria associated with *C. cyphergaster* or the mechanisms that affect their interactions. The rather varied diet of *C. cyphergaster*, for example (as compared to other termites), can influence their interactions with gut microorganisms, as they feed on lichens, deadwood, the surface of the bark of live trees, Cactaceae, and leaf-litter in their shrubby caatinga habitat [21,38]. Another interesting application of genetics is the study of phenotypic plasticity, which displays considerable variation among samples from multiple locations. Transcriptomic and Functional analyses have been used to study phenotypic plasticity, as with *Macrotermes barneyi* [39].

*Constrictotermes cyphergaster* shows great sensitivity to environmental conditions, preferring to nest mainly in locations without any strong anthropic presence. The species requires special above ground conditions and supporting trees for successful establishment [25]. It is interesting, however, that most of the research that has generated information related to this species (which is very sensitive to disturbance) has been carried out under laboratory conditions (Table 1). We understand the importance of research with controlled variables, such as temperature and moisture, but nonetheless encourage more field observations that can be merged with laboratory observations. Studies carried out in the Cerrado areas demonstrated fewer field experiments than those carried out in the Caatinga domain, resulting in gaps in our knowledge in the Cerrado areas concerning interactions with termitophiles during foraging (Figure 6) [40,41], lichen consumption (Figure 7), the identification of the assemblage composition of the lichens consumed, and the possible relationships between lichens and nests [21,42,43].

Lichen consumption is also a fertile research subject, as little is currently known about many aspects of this rather rare behavior among termites, with significant gaps in our knowledge of the behavior and the nutritional ecology of this species [23,38,42,44] that deserve investigation. (1) If lichens are important dietary items of this species [42], how does *C. cyphergaster* behave when deprived of that resource? Do they search for alternative dietary complements? (2) Are all resources consumed purely because of nutritional demands? Is lichen consumption linked to other necessities, such as parasite and pathogen control?

One of the most intriguing phenomena surrounding the ecology of social insects is the presence of an associated fauna. Termite colonies harbor several guests from recurrent lineages (i.e., Coleoptera, Collembola, Diplopoda, etc.) [45], as well as inquilines (termite species that inhabit the nests of other termites) [29]; these host–guest relationships have been the target of 12 studies. One question that is currently being investigated, but still requires additional studies, is the positive relationship between nest volume and cohabitation (which appears to be linked to nest defense strategies) [46].

The *Inquilinitermes*-*Constrictotermes* model has so far been the most studied case of inquilinism among termites [29,46,47]. Cunha et al. [48] established the starting point for those studies by observing that *Inquilinitermes* colonies displayed a correlational preference for establishing themselves in nests with a 33 cm longitudinal diameter threshold. Building on that, Cristaldo et al. [29] concluded that inquilines are present only if the nest has a volume greater than 13.6 L, which appears to be correlated with the accumulation of “black mass” (feces and other materials that form a humid black to fill in parts of *C. cyphergaster* nests). Both texts provided the basic information needed to design methods to test inquilinism hypotheses in the model.

Although *C. cyphergaster* represents a well-studied model (as compared to other termite species), there are no signs yet that research production has peaked concerning termitophiles. Zilberman [49] reported the existence of the anomalous *Corotoca hitchensi* (Coleoptera: Staphylinidae), a species described based on only a single specimen collected by the late termitologist Renato L. Araújo. Not much is known, however, regarding the habits of any species of *Spirachtha* (sister to *Corotoca)*, with one species associated with *C. cyphergaster* (*S. eurymedusa*). Given the viviparity reported by Schiødte [50] it could be expected that they would demonstrate behaviors similar to *Corotoca* during foraging events (giving birth to motile larva), but with some differences, as female specimens of *Spirachtha* rarely carry larvae, in contrast to the continuous pregnancies of *Corotoca* (Zilberman, personal communication). It is possible that they do not reproduce as frequently, or that they do not adapt well to the varied environments found throughout their host territory.

One of the gaps in our knowledge concerning this termite relates to the part of the nest that contacts the soil [51]. How does the soil respond to the presence of termites? Why do these termites maintain links to the underground portion? Which physical, chemical, and biological characteristics of the soil are modified by these constructions and by the presence of termites? Bezerra-Gusmão et al. [18] highlighted the importance of *C. cyphegaster* tree nests to the carbon cycle, and its accumulation in the Caatinga areas. Considering the underground nest portions, we are able to formulate some hypotheses concerning how these termites influence the soil.

As in other ecosystems, soils are hierarchical systems, with internal processes occurring at each level of organization [52]. Termite-driven modifications of the soil microbiota can alter the ecosystem services of microorganisms, and one of these services is the interaction of bacteria/fungi with plant roots [53]. The link between *C. cyphergaster* nests and the roots of their supporting trees would presumably influence these interactions. It is likely, for example, that these termites can transport fungal spores through the different soil horizon layers, and that those fungi could interact with more plant roots, modify the soil, and alter the microhabitats of all of the microorganisms present. As *C. cyphergaster* trails have mean lengths of approximately 18 m, those termites will interact with microorganisms on the ground surface, tree bark, and on dead wood over large distances (more than 170 m^2^ around the nest) [38].

## 5. Conclusions

Representative species (such as *C. cyphergaster*) are essential to ecology and conservation, although we agree that biological processes may change at global scales, and case studies may not establish standards for all populations of a given taxon. Some of the fundamental ecological questions concerning populations, however, pointed by Sutherland et al. [54] (such as “How do species and population traits and landscape configuration interact to determine realized dispersal distances?”) can be efficiently tested using *C. cyphergaster*. As that species builds its nests in trees and soil, another good question is “How does below ground biodiversity affect above ground biodiversity and *vice versa*?” Our research group is presently carrying out directed studies of that nature, but many additional questions can be tested using such a model. As a final note, we suggest strengthening links between researchers and research groups that could facilitate the collection of information regarding this species, link the different biomes in which they are found, and facilitate the production of generalizations concerning the group.

## Figures and Tables

**Figure 1 insects-13-00704-f001:**
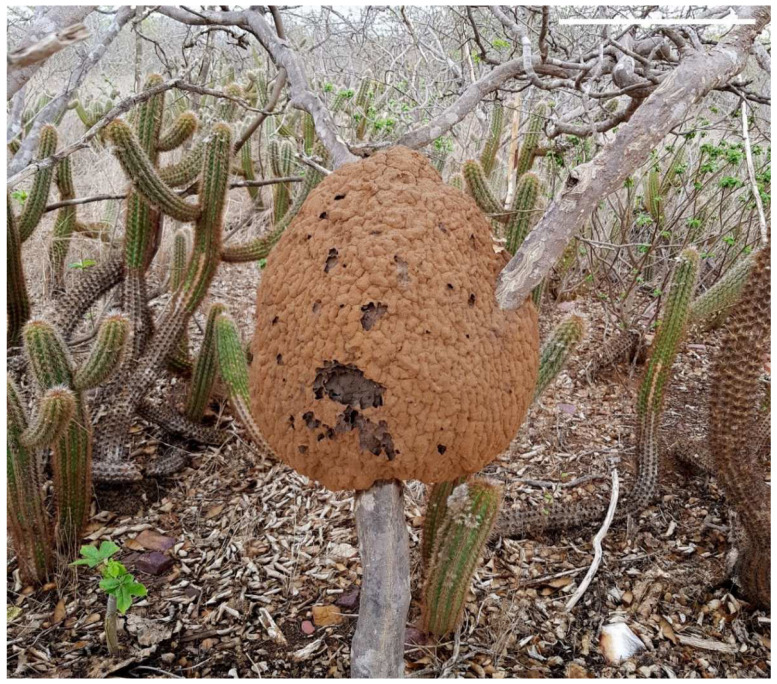
Nest of Constrictotermes cyphergaster (Blattodea, Termitidae) in a Caatinga forest, Northeastern, Brazil. Scale bar on top right = 30 cm.

**Figure 2 insects-13-00704-f002:**
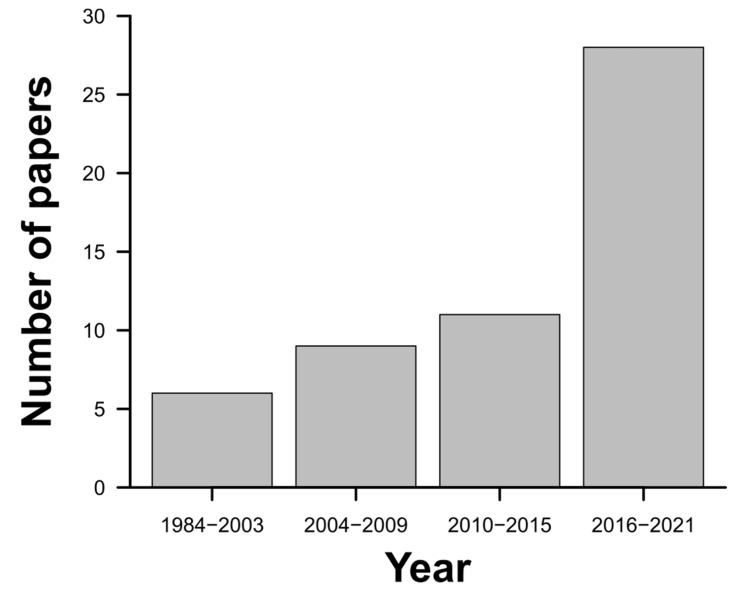
Distribution of the 54 papers evaluated in the present study over the time (years).

**Figure 3 insects-13-00704-f003:**
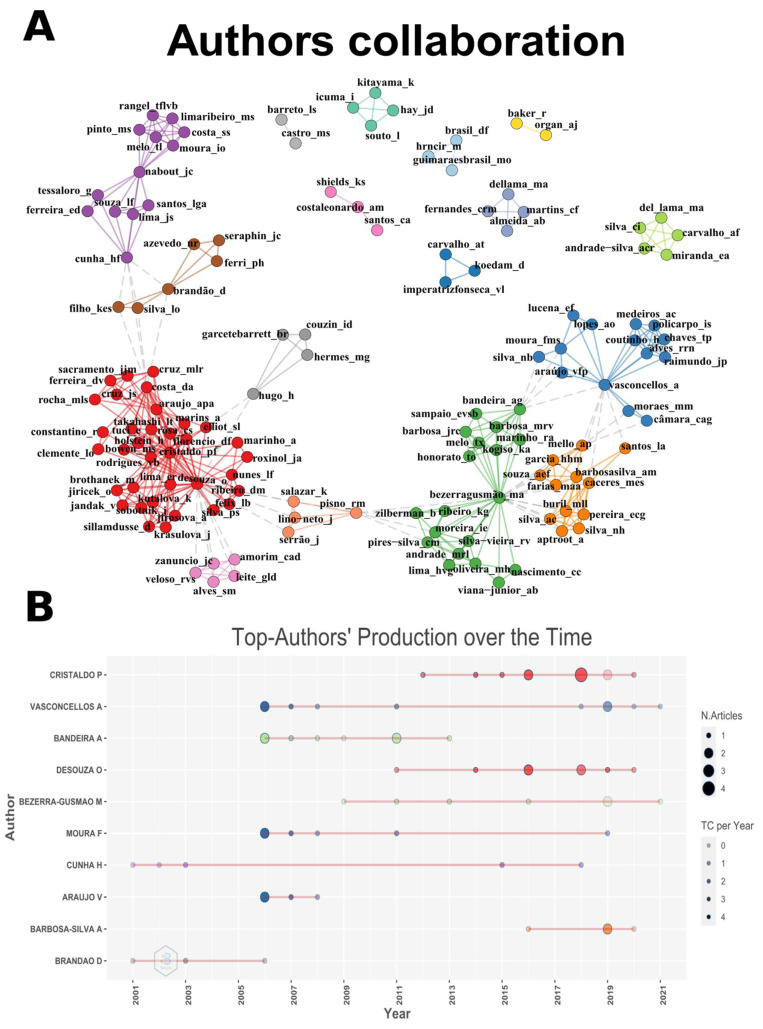
(**A**) Authors collaboration network. Different colors represent clusters formed according to links between authors. (**B**) Papers production of top authors over time (Years). Colors are linked to the colors of webs’ clusters. N. articles is the number of articles published.

**Figure 4 insects-13-00704-f004:**
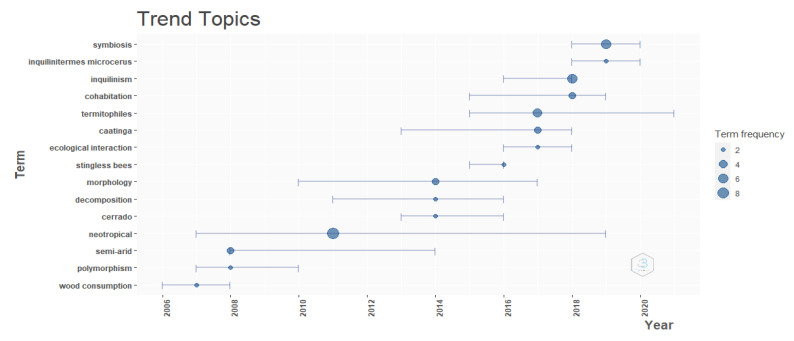
Trend topics observed in keywords of the 54 works about *Constrictotermes cyphergaster* utilized in present study. The plot shows the frequency of terms (dots) over years and in which years terms were present (whiskers).

**Figure 5 insects-13-00704-f005:**
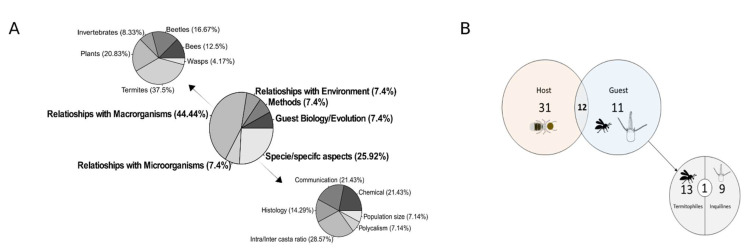
Analysis of content present in the papers analyzed. (**A**) The central pizza plot shows the fractions of the total number of papers (*n* = 54). The top left piechart plot shows the divisions of papers produced inside the “Relationships with macroorganisms” category (*n* = 24). The bottom right piechart shows the divisions of papers produced inside the “Species/specific aspects” category (*n* = 14). (**B**) Number of papers produced which aimed *C. cyphergaster* (host), its guest (termitophiles/inquilines), and both (host/guest).

**Figure 6 insects-13-00704-f006:**
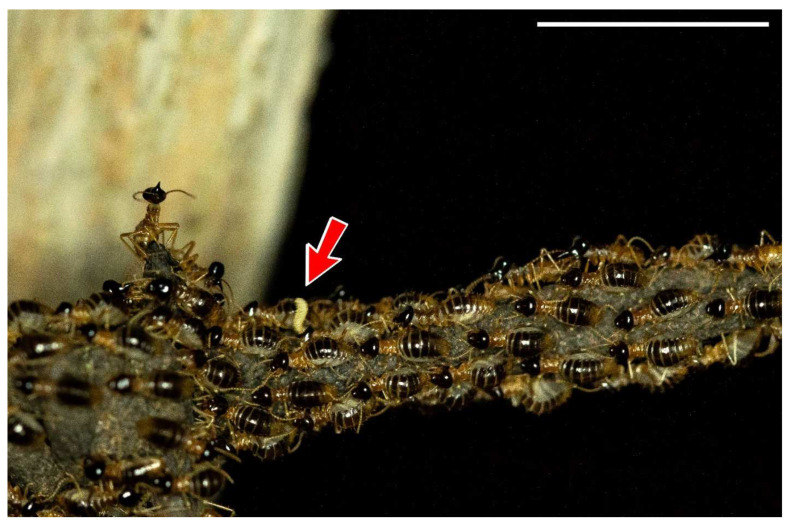
Foraging trail of *Constrictotermes cyphergaster* (Blattodea, Termitidae). The arrow shows a *Corotoca* sp (beetle that lives only inside the nests of *C. cyphergaster* in the semiarid areas of South America) larva being carried by a termite worker. Scale bar on the top right = 1 cm.

**Figure 7 insects-13-00704-f007:**
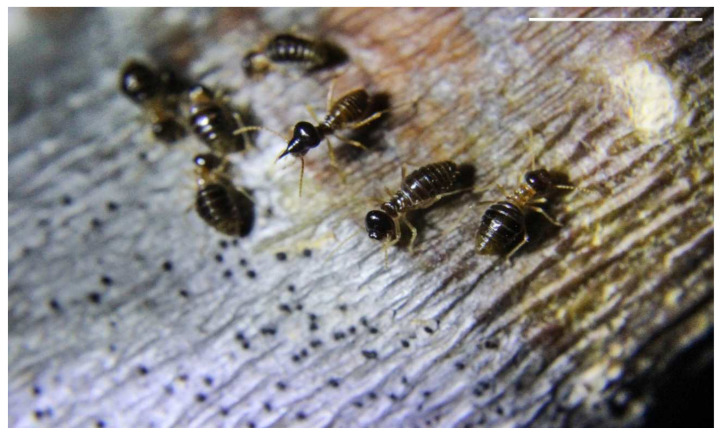
Individuals of *Constrictotermes cyphergaster* (Blattodea, Termitidae) foraging on lichen. Scale bar = 0.5 cm.

**Table 1 insects-13-00704-t001:** Number and % of papers produced with information generated in field, laboratory, and both. “Description” reflects the description of the sampling site described by authors in the articles.

Conduction	No.	%	Description
Laboratory	25	46.30	Caatinga (7), Cerrado (10), undescribed (8)
Field	21	38.89	Caatinga (14), Cerrado (6), undescribed (1)
Both	8	14.81	Caatinga (4), Cerrado (4), undescribed (0)

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
