# Peer review of "The Impact of Constrictotermes cyphergaster (Termitidae: Nasutitermitinae) Termites on Semiarid Ecosystems in Brazil: A Review of Current Research"

_insects, 2022, doi:10.3390/insects13080704_

Round 1

Reviewer 1 Report

The manuscript has been improved after the revision.  I noted one minor correction in line 92 change "half" to "mid".

Author Response

Dear reviewer, thank you very much for your revision. The English of the manuscript was submitted to an English editing by a native speaker to improve the quality of the manuscript.

Reviewer 2 Report

I have some editing comments in the attached pdf.  However, this manuscript is not improved from the last time I reviewed it.  The English still needs serious editing, and it cannot be published as it is.  Aside from the writing, the methods of finding references are inadequate for a "review" paper.  The authors have pulled papers from search engines, but not used those papers to find older references.  So what we have here is a list of what search engines can find.  And a bunch of statistics on what search engines found, which might not be all that valuable.

Author Response

Dear reviewer, thank you very much for your revision. The English of the manuscript was submitted to an English editing by a native speaker to improve the quality of the manuscript.

We used the databases for the scientometric analyses, but we discussed information of old papers too. You can find in references papers from 1853 and other years of past centuries, encompassing a rich range of information about the species.

The prism protocol is used in reviews of other groups and organisms. It is a method of papers published in high impact journals and have no problem with content, if researchers merge it with their experience to know if databases can hold the maximum of information about the organism that they are studying.

Line 4: “This needs to be adjusted.  Either remove the "ly" or rewrite this part of the title.”

The title was changed

Line 16: “Please rewrite.  This does not make any sense.”

The sentence and the simple summary passed by a review and were modified.

Line 20: play important roles

Okay. Modified.

Line 22: “I would tell the authors how to fix this, if I could figure out what this part of the sentence means”

The sentence was removed.

Line 24: “species”

Okay. Modified.

Round 2

Reviewer 2 Report

While the English has been improved since I last read it, there are still a few minor corrections to be made.

More importantly is the review itself.  It seems to serve less of a means of informing the reader of the content of papers in this area, and more of a game of word and author association.  Review papers are meant to be the former.

Advice: get rid of all figures that are not photographs, as well as the associated text.  Replace it by telling the reader what these papers were about in depth, not just a shallow scanning of the papers.

Author Response

Dear reviewer,

Thank you for your comments. We agree that the review is important. The use of scientometrics data is just to show the increase of interest and publications concerning the termite. You can find information of all the papers in the discussion. We discussed all topics that we found in the content analyses.

This manuscript is a resubmission of an earlier submission. The following is a list of the peer review reports and author responses from that submission.

Round 1

Reviewer 1 Report

It is the systematic review of the species Constrictotermes cyphergaster in open vegetation in Brazil.

There is a lot of research about this species and this MS is the first about a synthesis of the knowledge.

This MS made a synthesis of all the research and publication done previously and also made a qualitative analysis of the results, with suggestions for complementary research.  The authors met the proposed objectives.

The references are appropriate.

Figure and tables are fine.

The MS is very interesting and the results are clear. I detected a few mistakes

Keywords: do not repeat words in the title, change "review" to "research gaps", for example

line 62: replace (Constantino, 1998) to [18]

line 194: replace whit to whith 

Reviewer 2 Report

The manuscript certainly contains a great deal of information on this species of termite and will definitely be of interest to other termite researchers.  My most serious concern is the need for the English grammar t be corrected.  It is very distracting to read unclear sentences.  The title would be more appropriate if it stated up front it was a review, i.e., something like: "A review of Constrictotermes ....".  Line 44 needs to mention carbon dioxide.  Line 92 should indicate what "TS" stands for.  Scheme 1: change "52 of studies duplicates" to "52 unique studies".  

Reviewer 3 Report

After reviewing this manuscript I feel that the English quality should keep it from publication.  However, there are other issues as well.

    1. This is not really a "review paper" in the traditional sense, it is instead a review of papers found in three literature databases.  As far as I can tell, no attempt was made to go further than the three internet databases that were searched to find the 54 publications that the rest of the paper is based upon.

     2.  Apparently the authors have just discovered R as a graphing tool.  Most of the graphs are inappropriate and seem to be included to make the paper seem more technical.  Figure 1 is somewhat useful, but Figures 2 and 3 should be deleted.  Figures 2 and 3 are misleading and do not provide any useful information - how often do certain authors work together and their publication records?  These questions do nothing more than pad the paper.